# Natural Biopolymers as Additional Tools for Cell Microencapsulation Applied to Cellular Therapy

**DOI:** 10.3390/polym14132641

**Published:** 2022-06-29

**Authors:** Liana Monteiro da Fonseca Cardoso, Tatiane Barreto, Jaciara Fernanda Gomes Gama, Luiz Anastacio Alves

**Affiliations:** Laboratory of Cellular Communication, Oswaldo Cruz Institute, Oswaldo Cruz Foundation, Brazil Avenue, 4365-Manguinhos, Rio de Janeiro 21045-900, Brazil; tatiane.barreto321@gmail.com (T.B.); bmd_gomes@yahoo.com.br (J.F.G.G.); alveslaa40@gmail.com (L.A.A.)

**Keywords:** graft rejection, natural polymers, microencapsulation, cell therapy

## Abstract

One of the limitations in organ, tissue or cellular transplantations is graft rejection. To minimize or prevent this, recipients must make use of immunosuppressive drugs (IS) throughout their entire lives. However, its continuous use generally causes several side effects. Although some IS dose reductions and withdrawal strategies have been employed, many patients do not adapt to these protocols and must return to conventional IS use. Therefore, many studies have been carried out to offer treatments that may avoid IS administration in the long term. A promising strategy is cellular microencapsulation. The possibility of microencapsulating cells originates from the opportunity to use biomaterials that mimic the extracellular matrix. This matrix acts as a support for cell adhesion and the syntheses of new extracellular matrix self-components followed by cell growth and survival. Furthermore, by involving the cells in a polymeric matrix, the matrix acts as an immunoprotective barrier, protecting cells against the recipient’s immune system while still allowing essential cell survival molecules to diffuse bilaterally through the polymer matrix pores. In addition, this matrix can be associated with IS, thus diminishing systemic side effects. In this context, this review will address the natural biomaterials currently in use and their importance in cell therapy.

## 1. Introduction

The treatments of many diseases through organ, cell and tissue transplantation are limited by graft rejection. In order to avoid early graft rejection, transplant patients must take immunosuppressive agents (IS) in the long term. However, several side effects including infections [1], renal toxicity, neurotoxicity [2,3], bone marrow suppression, malignancies [4] and metabolic disorders are associated with the chronic use of IS drugs [5]. In this context, alternative strategies to replace or reduce IS agent administration protocols have been put forth, such as a combination of agents with different mechanisms of action and toxicity profiles, dose reduction or complete IS withdrawal. Many patients, however, do not adapt to these protocols and must return to conventional IS use. Therapy must be individualized, and additional preventive measures should be taken by patients who present particular risk factors or are predisposed to certain adverse effects. This makes it even more difficult to standardize a single less invasive protocol [6,7,8,9,10].

In addition, the shortage of available organs for transplantation in relation to the number of patients in need, as well as the limitation of cellular and tissue sources for transplantation, impose a major obstacle to regenerative therapy and the treatment of numerous diseases. These problems could be addressed and even solved by employing allogeneic or xenogeneic cell sources. However, graft rejection issues and the need for IS drugs would remain unsolved. Currently, a promising alternative to graft non-rejection has been studied and, in some cases, successfully performed for the treatment of some diseases—namely, cell encapsulation—based on cell immobilization within a semi-permeable hydrogel matrix [11]. Cell immobilization in polymer-based hydrogels was first proposed in 1933 by Bisceglie [12], who demonstrated that insulin-producing cells remained viable and metabolically active after immobilization. Three decades later, Chang proposed the use of semipermeable membranes as immune-isolating devices and introduced the term “artificial cells” to define the cell microencapsulation concept [13]. The semi-permeable membrane protects internal cells from both mechanical stress and the host’s immune system while allowing for bidirectional nutrient and oxygen diffusion and waste disposal, viability and metabolic functionality for subsequent therapeutic applications. Additionally, immunosuppressants or immunomodulatory chemokines, such as CXCL12, can be incorporated into the alginate to further decrease or prevent the recipient’s immune response, thus avoiding the need for systemic immunosuppression. These factors thus enable the application of therapy cell encapsulation (Figure 1) [14]. This is an important advantage that may lead to a reduction or even the absence of continuous IS administration, an important barrier to be considered in successful organ transplantation due to several side effects.

Cell encapsulation instead of therapeutic product encapsulation leads to longer delivery times, as cells continuously release the encapsulated products. Moreover, cell encapsulation allows for the transplantation of non-human cells, which may be considered an alternative to the limited supply of donor tissues [15,16]. In addition, genetically modified cells could also be immobilized to express any desired protein *in vivo* without host genome modifications. Cell immobilization displays an important advantage compared to protein encapsulation, allowing for the sustained and controlled delivery of ‘de novo’-produced therapeutic products at constant rates, leading to physiological concentrations [15]. The versatility of this approach has stimulated its use in the treatment of numerous medical diseases including diabetes [17,18], cancer [19], central nervous system diseases [20], heart diseases [21] and endocrinological disorders, among others [22]. In addition, immunosuppressive drugs may also be incorporated in hydrogels as an additional mechanism to avoid immune-mediated rejection, leading, in principle, only to local transplanted encapsulated cell effects. In this context, microencapsulation is widely investigated as a safe path to β-cell replacement by islet transplantation for the treatment of type 1 diabetes [23,24]. Nonetheless, inflammatory foreign body responses leading to pericapsular fibrotic overgrowth often causes microencapsulated islet-cell death and graft failure. Therefore, glycemic correction is only achieved for short periods. Stem cell-derived β-cells (SC-β cells), functional β-like cells from human pluripotent stem cells, when co-encapsulated with CXCL12, an immunomodulatory chemokine, showed enhanced insulin secretion in diabetic mice and accelerated the normalization of hyperglycemia. Additionally, SC-β cells co-encapsulated with CXCL12 evaded the pericapsular fibrotic response, resulting in long-term functional competence and glycemic correction without systemic immunosuppression in immunocompetent mice [25,26].

Many different materials are employed to encapsulate cells, many naturally derived from hydrogel-forming polymers. In this regard, hydrogels are among the most promising biomaterials for the recreation of native extracellular matrix (ECM) properties due to their high-water content, biological compatibility and moldability [27]. Several types of hydrogels obtained from natural polymers, synthetic polymers and co-polymers displaying optimized physical and chemical properties have been developed for regenerative medicine [28,29].

Hydrogels derived from natural materials are similar to the ECM of many human tissues, lacking proteins such as laminin and fibronectin, among others, that are vital for several cellular functions. These gels are made of polymers similar to the biological macromolecules engineered by nature to perform specific functions in a demanding environment. Some of them are abundant (e.g., from marine sources) [30] and can often be processed under conditions compatible with cell survival [31].

In addition to their application to advances in cell therapies, some biomaterials are already used in other areas and can be found commercially. Alginate hydrogels have been particularly attractive in wound healing and can be found in the form of dressings. Alginate wound dressings maintain a physiologically moist microenvironment, minimize bacterial infection at the wound site, and facilitate wound healing [32]. Alginate dressings are typically produced by the ionic cross-linking of an alginate solution with calcium ions to form a gel, followed by processing to form freeze-dried porous sheets (i.e., foam) and fibrous non-woven dressings. Alginate dressings in the dry form absorb wound fluid to re-gel, and the gels can then supply water to a dry wound, maintaining a physiologically moist microenvironment and minimizing bacterial infection at the wound site. These functions can also promote granulation tissue formation, rapid epithelialization and healing [33,34]. Various alginate dressings, including Algicell™ (Derma Sciences, Princeton, NJ, USA) AlgiSite M™ (Smith & Nephew, London, UK), Comfeel Plus™ (Coloplast, Humlebaek, Denmark), Kaltostat™ (ConvaTec, Reading, UK), Sorbsan™ (UDL Laboratories, Van Nuys, CA, USA) and Tegagen™ (3M Healthcare, Saint Paul, MN, USA), are commercially available [34] Alginate-based compounds have also been used in the treatment of gastroesophageal reflux disease (GERD) symptoms. In the United States, they are typically sold under the brand name Gaviscon in both tablet and liquid formulations, which are available without a prescription [35]. Alginic acid is listed as an inactive ingredient. The brand name Gaviscon, however, is used to market alginate-based therapies in a number of other countries including Canada and throughout Europe. Formulations such as ‘Gaviscon Acid Breakthrough’ in Canada list alginic acid as an active ingredient, similar to ‘Gaviscon Advance’ in the United Kingdom [36,37]. Furthermore, pharmaceutical products based on alginates include: rectal agents (Natalsid suppositories [38]) used for chronic hemorrhoids, proctitis and chronic anal fissures after rectal surgery; periodontal agents (Progenix putty, Progenix plus injection [39]) used for bone defects. In addition, Emdogain gel [40,41] is used for intraosseous defects and defects of mandibular furcation with minimal atrophy of the interdental bone, and agents applied arthroscopically (ChondroArt 3D injection) are used in degenerative diseases of the joints and spine [42].

Hyaluronic acid is one of the most efficient and safe ingredients used frequently in cosmetics. Some formulations containing HA are already available on the market, with a large experience in their use. Fresh (USA), Deciem (Canada), Apivita (Greece) and Farmec (Romania) are some examples of products [43]. Hyaluronic acid and its derivates are incorporated in a multitude of cosmetic products for eye contour, lips, facial, and neck care, anti-cellulite body care or cosmetic color conditioning in different cosmetic categories: creams, lotions, serums and masks [44,45].

This review mainly discusses natural biopolymers, whose sources and chemical structure are illustrated in Figure 2, and their importance for cell therapy.

## 2. Natural Biopolymers

### 2.1. Chitin and Chitosan

Chitin and chitosan belong to the biopolymer family composed of N-acetyl-D-glucosamine and D-glucosamine, which are linked by β(1→4) [46]. Chitin is the second most abundant natural polysaccharide in the marine environment, and its main source is the exoskeleton of molluscs, crustaceans, insects and fungi cell walls [47]. Three different chitin polymorphs are noted, namely, in the form of alternating leaves with parallel and antiparallel chains, organized into polysaccharide chains arranged in parallel by weak intermolecular forces and arranged in two polysaccharide chains in the form of parallel and antiparallel sheets [48]. Although chitin is a biocompatible, biodegradable compound with a high mechanical strength, it presents an important limitation in clinical applications, namely, low solubility. In order to improve chitin solubility, chemical or enzymatic processes and deacetylation are used to obtain a more soluble and bioavailable compound, termed chitosan [49], whose applicability has been demonstrated in the textile, food, medicinal and agricultural fields [50].

The nanoparticles of this biopolymer have been applied in the pharmaceutical and biomedical industries, and it displays an important role as a drug carrier—for example, against Mycobacterium bovis (M. bovis) infection [51] and as a hormone carrier, as in the treatment of diabetes mellitus-type 1 [52]. In addition, chitosan, a cationic low-toxicity and immunogenicity compound exhibiting biocompatibility, is also an excellent delivery system for plasmid DNA (pDNA), oligonucleotides and interfering RNA (siRNA). Its positive charge throughout the polymer chain forms a complex with negatively charged nucleic acid, allowing for gene delivery to the desired location [53]. Thus, it has been employed as a gene carrier agent, as viral vectors can, in some cases, cause immune responses, toxicity, inflammation and immunogenicity during gene therapy [54].

Chitosan has been used in cell therapy as a three-dimensional polymer scaffold for cell growth. Alinejad and coworkers observed that the microencapsulation of mammalian cells in polymeric shades of chitosan by the emulsification technique resulted in resistance to compressive forces, thus comprising a suitable microenvironment for *in vitro* cell survival [55]. In another study, chitosan associated with Poly (ε-caprolactone, PLC), a synthetic biodegradable polyester compound, played an important role in bioengineering, namely, in the repair of injured bladders of murine models. In this case, stem cells derived from adipose tissue were cultivated in this matrix and participated in the anatomical reconstruction of the bladder, assisting in the development of extra layers of smooth muscle tissue. In addition, extra tissue layers were formed outside of the three-dimensional chitosan–PLC structure, suggesting that the three-dimensional polymeric structure served as a support [56].

In one assessment of myocardial regeneration, chitosan was associated with collagen in order to improve its polymeric structure, leading to better cell differentiation and adherence, as well as angiogenesis induction. Thus, this structure offers a cell growth scaffold and promotes endothelial cell recruitment. In addition, chitosan favored the blood vessel induction, related to the interaction between its amine groups and glycosaminoglycan (GAGs) [57]. Chitosan associated with β-glycerol phosphate has been reported as improving cell survival, as well as cell proliferation and migration, leading to the *in situ* release of therapeutic molecules and also exhibiting thermosensitivity, an important characteristic associated with this modification. Thus, an angiogenesis function resulting in ameliorated peripheral arterial disease is noted [58]. Another important association is observed between chitosan and immobilized growth factors, such as IGF-1. In this case, chitosan was able to prevent programmed cell death, as its structure provides an essential microenvironment for cell adherence, proliferation and anti-apoptotic and pro-angiogenesis action leading to cell protection and kidney recovery and regeneration in an animal model [59]. Mesenchymal stem cells encapsulated with chitosan linked to celastrol, an antioxidant compound extracted from plants, have been associated with increased cell survival and viability, as well as an increase in cytokine release and angiogenesis, improving tissue regeneration. This strategy may be applied to several biopolymers, cells and molecule encapsulation [60]. Yang Liu and co-workers demonstrated that a chitosan support leads to inflammatory response inhibition and improves angiogenesis after acute myocardial infarction in mice [61]. In addition, chitosan associated with peptides induces cell migration and adhesion (e.g., RGD), inhibiting keloid formation, as well as fibrotic collagen impairment after subcutaneous injection in mice and *ex vivo* analyses [62]. It is important to note that chitosan structure modifications improve tissue regeneration, which is also observed for chitin, although this chitosan precursor is less stable and bioavailable. When it is in a solution with sodium hydroxide and urea, chitin can enhance other biomaterial formations, such as carboxymetilchitin, a non-cytotoxic and biodegradable compound displaying high potential in 3D cell culture and clinical applications in tissue regeneration [63].

Thus, chitosan displays various applications. However, its use is widely studied in association with other polymers obtained naturally or synthetically. For pharmaceutical and biomedical applications, chitosan must first undergo several chemical modifications to improve its specificity and bioavailability [64]. It is important to highlight that the use of polymers that mimic the extracellular matrix as closely as possible is recommended in cell therapy, aiming at cell survival in this environment.

### 2.2. Hyaluronic Acid

Hyaluronic acid (HA) is a biopolymer composed of repeating disaccharides, namely, D-guluronic and N-acetyl-D-glucosamine, by linked β(1-4) and β(1-3) along its polymeric chain. It is a Glycosaminoglycan (GAG) family polymer found in all vertebrates, mainly in the extracellular matrix of connective tissues [65]. This compound exhibits high biocompatibility and biodegradability due to its ubiquitous distribution in the human body and plays an important role in cell division, differentiation and migration [66]. In addition, it is associated with angiogenesis [67], would healing [68], skin rejuvenation [69] and injury tissue regeneration [70,71], and it can be employed in drug delivery [72]. However, beneficial biological functions depend directly on its molecular weight, where high molecular weight HAs (>5 MDa) may induce anti-angiogenic and immunosuppressive events and low molecular weight HAs (6–20 kDa) can trigger pro-inflammatory processes and angiogenesis and often induce the gene expression of heat-shock proteins. HA polymers between 20 kDa and 1 MDa are commonly used in injured tissues healing and regeneration [73]. 

This biopolymer can be employed for several proposes—for example, as a drug carrier, when applied to the treatment of psoriasis, HA facilitated the dissipation of the drug Methotrexate under the skin, considerably improving cutaneous inflammation [74]. In addition, HA has also been used on chitosan nanoparticle surfaces for drug delivery as an alternative therapy for bronchiolitis obliterans syndrome, exhibiting low toxicity and facilitating the internalization of this complex by cell plasma membranes, consequently favoring drug release in the intracellular microenvironment [75]. Concerning tumor cell lines, HA promoted nanoparticle internalization by binding to receptors, influencing the release of antitumor drugs in the intracellular environment. The formed chitosan–HA nanoparticle complex induced the generation of Reactive Oxygen Species (ROS), damaging the mitochondria membrane integrity and inducing the apoptotic cell pathway [76].

Carboxyl and hydroxyl groups found in the HA structure confer hydrophilicity, facilitating cell adhesion and absorption in a microenvironment containing high serum protein and leukocytes content, so HA comprises an essential polymer for tissue recovery. In addition, its other groups playing a role in hemostasis, local inflammation regulation and healing process stimulation can be easily chemically modified [77]. This was observed in rats with palatal wounds, where HA participated in injury repair and decreased inflammatory reactions up to seven days, with fibroblast expansion resulting in increased collagen deposition [78]. Concerning skin healing, tissue regeneration was observed when HA was associated with Poloxamer 407, a non-ionic and non-toxic copolymer that plays an important role against bacterial activity in injuries caused by skin excision on the back of adult male rats under an adequate oxygen supply. This role is greater than the basic fibroblast growth factor (bFGF) widely applied in this model [79]. Thus, HA plays an important role in wound healing, decreasing inflammation, increasing local vascularization and stimulating collagen synthesis.

When associated with sodium alginate, a polymer from a natural source used as a polymeric matrix for the encapsulation of insulin-producing cells isolated from rats, HA was able to increase the survival of microencapsulated cells. The alginate–HA matrix decreased the apoptosis of encapsulated cells, leading to increased cell viability and the preservation of insulin secretion capacity [80].

HA has been also associated with Cucurbit [6]uril (CB[6]-HA), macrociclic molecules and 1,6-diaminohexane (DAH-HA) for therapeutic approaches employing mesenchymal stem cells, with increased cell survival under both conditions (CB[6]-HA and DAH-HA). It also decreases melanoma tumor growth in C57BL/6J mice, suggesting a potential use as an anti-cancer therapy [67,81]. Stem-cell encapsulation with co-acervates promotes cell spreading and attachment on target tissues, also increasing HIF-1α, a transcriptional cell regulator, under hypoxia. In addition, angiogenesis and cell proliferation were observed without inflammatory responses [82]. Another key modification that improves the therapeutic function of this biopolymer comprises the incorporation of polyethylene glycol into the HA matrix (PEG-HA), creating an important microenvironment for cell encapsulation, as the polymeric structure has been reported as capable of inducing canine pancreatic islet survival after transplantation. In addition, blood glucose returned to normal levels in diabetic NOD/SCID mice long-term after microencapsulated islet transplantation. On the other hand, the association between HA and methyl methacrylate (ME-HA) did not affect blood glucose levels in another study [83] These data indicate that associations between HA and PEG induce normal blood glucose and facilitate suitable cell proliferation.

In line with this idea, the HA polymer also displays a wide applicability as a three-dimensional support for cell growth. However, despite being widely used, HA is usually associated with other biopolymers to prevent degradation by body enzymes, such as hyaluronidase, which could generate Nitric Oxide Syntase (NOS) and ROS, which consequently induce inflammatory responses [84,85].

### 2.3. Agarose

Agarose is another polysaccharide applied in tissue bioengineering and is obtained mainly from the marine alga Ahnfeltia plicata, belonging to the Ahnfeltiaceae family [86]. Its structure is composed of disaccharide repeats of β-D-galactose and 3,6-anhydro-α-l-galactopyranose [87], creating double helices in the polymer chain and aggregating, creating a three-dimensional matrix [88]. Due to this three-dimensional polymeric network and its similarity to the extracellular matrix, agarose can be used in the regeneration and recovery of injured tissues, especially when associated with other extracellular matrix components, such as collagen, laminin and fibronectin [89].

Agarose has been used by molecular biologists in the analysis and separation of DNA fragments by gel electrophoresis and chromatography since the 1970s, and it has been used as a substrate for bacterial growth since the 19th century [90]. However, only in the last few decades has it been applied as a support for cell growth in 96-well plates to induce spheroid formation. This is due to the fact that agarose is a non-toxic compound with cell growth biocompatibility, consequently inducing cell aggregation and minimizing the inflammatory response due to its low cell adhesion capacity [91]. When associated with other polymeric components, such as polydopamine (PDA) extracted from mussels, the formed polymeric matrix exhibits low cytotoxicity *in vitro* experiments with fibroblast tumor cells. *In vivo*, the polymeric matrix exhibits a faster degradation rate in a murine model when compared to an *in vitro* model, suggesting an important role as cell growth and development support, as expected [92].

Although agarose is biocompatible and biodegradable, some factors affect its use in bioengineering, such as a high critical solution temperature, i.e., the highest temperature at which phase separation between two liquids occurs. Thus, more thermosensible drugs do not resist the procedure. The agarose polymer also displays a low degradation rate and a low adsorption capacity for hydrophobic drugs, as well as a slow desorption of certain drugs, which could compromise its role in the drug delivery [93]. It is important to highlight the urgent need for more research on the drug delivery, as the Food and Drug Administration (FDA) has not approved this compound due to poor evidence in this regard.

### 2.4. Collagen

Collagen is one of the most abundant extracellular matrix proteins, providing stability to both tissues and organs, molecular signal recognition and the formation of three-dimensional structures as a support for cell growth. It also displays controllable mechanical functions and biodegradability, resulting in tissue structural integrity [94]. A variety of collagens exist in the body, with 26 types described so far, classified as fibril-forming collagens (types I, II, III, V and XI), fibril-associated collagens (types IX, XII, XIV, XIX, XX and XXI), network-forming collagens (types VIII and X), anchoring fibrils (type VII) and transmembrane (types XIII and XVII) and basement membrane (type IV) collagens [95]. Despite the diversity of collagens, type I collagen is the most applied in tissue engineering, as it is easy to extract and displays high adaptability [96]. Thus, it comprises a natural source material used in tissue bioengineering for drug delivery, skin regeneration and bone tissue, tendon and cartilage regeneration [97], mainly due to its biodegradable, biocompatible and low antigenicity properties [98].

In tissue bioengineering, collagen has been used as a support for the delivery of human corneal stromal stem cells. However, that study highlighted a limiting property of collagen, namely, its low mechanical stability and greater deformation probability. In order to minimize this condition, the authors performed the compression of the collagen polymer followed by cell association in its polymeric scaffold. The complex formed between the collagen and the cells presented a high tensile strength with sufficient malleability to shape the injured corneal surface, in addition to increasing cell viability in this microenvironment and post-freezing regeneration of the injured tissue [99]. In this context, another study employed type I collagen associated with agarose to microencapsulate mesenchymal cells. The addition of collagen in the microcapsules improved the microcapsule biodegradability and cardiac function due to an increase in the left ventricular ejection fraction in contractility. It also led to scar reduction and an increase in the tissue capillarity [100]. This suggests that collagen is a therapeutic alternative to treat myocardial infarction.

Collagen has also been applied as a bioink in 3D bioprinters and as a three-dimensional scaffold composed of collagen extracted from rats, displaying low cytotoxicity and metabolic cell activity maintenance. However, the main limitation concerning the use of this biomaterial is its low viscosity and low solubility at room temperature, making the bioprinting procedure a challenge [101]. Collagens extracted from human tissues have also been used as bioinks in 3D printers, aiming at the characterization of their physicochemical properties and possible bioink application. Thus, this natural human biopolymer presents relevant properties for tissue bioengineering, such as gelation time, biomaterial purity, structural integrity and stable compressive mechanical properties [102].

A proof of concept recently published in 2018 produced HA-based hydrogels, in which type I collagen was incorporated to promote the cell survival of human adipose stem cells (hASCs), potentially comprising a therapeutic alternative for the treatment of corneal blindness. Collagen was required to link the cells into the matrix, significantly improving cell survival and resulting in higher metabolic activity. In addition, this protocol was determined as an efficient possibility in a corneal tissue culture model due to cellular integration and epithelial growth [103]. Similarly, the association between HA and collagen exhibits high potential for the treatment of diabetic mice, as it positively modulated glucose for 4 weeks [104]. Thus, this method is promising as a potential alternative therapy in the treatment of diabetes due to glycemia normalization and the decreased immune response against the applied capsules.

However, further studies should be carried out before using collagen as a bioink in the clinical practice, as the viscosity of this biomaterial can affect cell viability due to increased shear forces at the time of bioprinting. Furthermore, collagen purity is a key characteristic that will dictate the success of the bioprinted material, as the risk of immune reactions is directly related to material impurity [105].

### 2.5. Sodium Alginate

One of the most employed biopolymers in cell therapy is sodium alginate, a compound extracted mainly from the cell wall of different species of brown marine algae such as Laminaria hyperborea, Macrocystis pyriferae and Ascophyllum nodosum [106]. There are mostly two main sources of alginates: algal sources and bacterial sources [107]. In the last century, alginates were extracted for the first time from the marine macroalgae, and then, after 80 years, a bacterial source, the mucous strain of Pseudomonas aeruginosa, was discovered, from which bacterial alginates were produced [108]. Bacterial alginates have also been derived from some *Pseudomonas sp*. and *Azotobacter sp* [109]. These two bacteria follow the highly comparable mechanisms for the biosynthesis of alginates. However, the bacterial alginates possess different characteristics and applications (e.g., the development of highly structured biofilms) [110,111]. This biopolymer comprises two polysaccharides, β-D-mannuronic acid (M) and α-L-guluronic acid (G), which are linked together by glycoside (1-4), structured in unbranched linear units that can form consecutive blocks of mannuronic acid (MM) or guluronic acid (GG) alone or alternate with each other (MGMG) [112]. Its composition characteristics will depend on the extraction source, which directly affects the alginate’s physical properties, molecular weight and conformational structure [113].

Alginate presents several relevant characteristics for its use in bioengineering—mainly, its biocompatibility, non-toxicity and non-antigenicity, as well as its inert nature, low financial costs and easy handling [114]. It has been applied in the drug delivery of proteins, vaccines and cells [115].

Recently, the drug Exemestane, widely employed in breast cancer treatments, was nanoencapsulated in alginate for *in vitro* release. The results confirmed the controlled release of the drug, which is effective for treatment *in vitro* conditions. Current treatments include surgery, immunotherapy, chemotherapy, targeted therapy, hormone therapy and radiation therapy and result in numerous adverse effects affecting the patient’s health [116], making the search for alternative therapies an emerging need. Another important study reported that liposomes, vesicles formed by phospholipids, encapsulated in an alginate matrix were transported directly and were able to release drugs directly to the colon cancer target, reaching higher drug concentrations in the tumor itself [117]. Thus, in addition to comprising an alternative to cancer treatment, one of the most frequent diseases worldwide with high mortality rates, the alginate biopolymer may also be used as a carrier agent for targeted drugs [118].

Kojayan and co-workers microencapsulated and cryopreserved pancreatic islets isolated from rats in order to analyze post-freezing cellular integrity, highlighting that the alginate polymer matrix was efficient in protecting the islets and, after transplantation, restored normal blood glucose levels [119]. This suggests an effective alternative treatment for another disorder that affects approximately 38.7 million people worldwide: diabetes mellitus [120]. This alginate polymer matrix has also been used to encapsulate chemokines to improve β-cell responses to insulin. The procedure led to blood glucose correction for over 150 days in immunocompetent diabetic mice, without the need for immunosuppressant administration and with a lower fibrotic process after transplantation when the cells were co-encapsulated with the chemokine. This is a proof of concept demonstrating the possibility of transplanting pancreatic β cells to treat patients with diabetes mellitus [25]. Alginate has also been used in cell encapsulation for insulin-producing cells (IPCs) derived from xenogeneic and allogeneic sources, aiming at the treatment of type I diabetes in mice and resulting in decreased blood glucose levels and encapsulated IPCs protection against immune responses. Thus, encapsulating IPCs from different sources may be a promising alternative in the treatment of diabetes, as a significant limitation in the supply of stem cells is noted [121]. This was also observed when encapsulating porcine pancreatic islets in alginate-chitosan-methacrylate glycol membranes, with the bioconstruct demonstrating high biocompatibility and long-term cell survival after transplantation [122].

In addition to diabetes, many other disorders are being studied with the aim of providing a viable, minimally invasive and economically cheaper alternative, promoting patient recovery in several conditions, such as in the recovery of neurological function after traumatic brain injury [123], acute myocardial infarction treatment [124] and acute liver failure [125]. Regarding the latter, a 2018 study reported promising results in humans, where liver cells from human neonates were isolated, encapsulated and cryopreserved in an alginate polymer to minimize the immunogenicity of the transplant. The alginate matrix also allows for the diffusion of nutrients and other molecules essential for the survival of transplanted primary hepatocytes, making IS unnecessary. This methodology could be effective in solving the main limitation of this procedure, comprising the immune rejection of these transplanted cells [126]. In this regard, in one study, allogeneic transplantation in the peritoneum of children with acute liver failure (ALF) from King’s College Hospital, London, United Kingdom, has been deemed safe, with only one non-transplant related death. In another evaluation, other patients exhibited improved hepatic and metabolic function and were able to wait for organ transplantation or restore their liver function. Chronic and granulomatous inflammation was observed in patients eligible for exploratory laparoscopy at 3–6 months of therapy, whose liver function was completely restored, suggesting that, after full organ function restoration, beads should be removed to avoid delayed immune responses [127]. These assessments are pioneer human studies of microencapsulated liver cells used to treat children, with the preliminary results significantly important for this research field. Although transplant effectiveness cannot be confirmed due to the need for further studies, the results are promising and may lead to a viable alternative liver transplantation therapy or comprise a bridge to patients awaiting organs.

Alginate has also recently applied to extracellular vesicle delivery from mesenchymal stem cells for the treatment of myocardial infarction. The vesicles were retained longer within the polymeric structure and released slowly, thus increasing pro-angiogenesis effects and contributing to the improvement of post-myocardial infarction cardiac function in rats [128]. Cardiomyocytes have also been microencapsulated in alginate-gelatin, with no observable cytotoxicity and increased cell viability and proliferation, demonstrating that this matrix is a viable structure for this treatment. However, an important limitation was evidenced: the low retention of encapsulated vesicles due to impaired blood flow, as well as cell retainment in the alginate polymer matrix [129]. In another study, adipose mesenchymal stem cells were also microencapsulated in alginate biopolymers in order to assess the role of microencapsulation on the assessed cells. The encapsulated cells exhibited over 70% cell viability and *in vitro* cell differentiation property maintenance. Meanwhile, when *in vivo*, these cells displayed a low local inflammatory response [130]. These data suggest that this biopolymer is an effective and safe way to microencapsulate cells for the treatment of various disorders. Recently, a study aimed at analyzing the delivery of microencapsulated mesenchymal stem cells in alginate-collagen-Dextran sulfate-agarose matrices in infarcted mice. The polymer microcapsules exhibited an optimized composition with good mechanical stability, not resulting in exacerbated fibrotic capsule reactions. In vivo, after transplantation, the microcapsules were slowly degraded over time, with gradual compound release from the encapsulated cells [129]. Cell microencapsulation may comprise a promising alternative to minimize the main limitation in the treatment of infarcted myocardium, i.e., cell permanence after transplantation due to semipermeability, favoring the entry of essential nutrients for cell survival and, at the same time, protecting against immunological action and preventing cell migration, thus allocating cells within the biopolymer matrix. Thus, the microencapsulating polymer matrix was proven suitable for cell delivery. In another assessment, mesenchymal stem cells encapsulated with modified alginate played an important immunomodulatory role, reducing immune responses and preventing the formation of pericapsular fibrotic tissue. The microcapsules demonstrated physical integrity after transplantation in mice and remained structurally viable, maintaining cell survival for up to 14 days, thus improving acute myocardial infarction [131]. An improvement in cardiac function was observed after stem cell transplantation from cardiovascular tissue encapsulated with alginate-poly-l-lysine membranes in an acute myocardial infarction model. The polymeric scaffold did not affect cell viability, with viable cells noted for up to 21 days. Growth factors were also produced by the cells, as angiogenesis was induced and the injured tissue was repaired [132]. Thus, the participation of alginate in several health fields is noteworthy, displaying promising results in the treatment of various disorders.

The advantage of using alginate, especially alginate commercially sold as ultrapure and with a high guluronic acid ratio, is a lower probability of immunogenic reactions and higher tissue recovery chances [133]. However, an inherent limitation of *in vivo* alginate procedures, depending on the time for which the biomaterial will remain inside the living organism, is a high degradation rate over time, although this possibility of degradation *in vivo* is great but can be solved by associating other polymers with the alginate matrix [134]. Therefore, complementary studies are required to clarify these questions and improve the use of this biopolymer in cell bioengineering, as well as to enable the use of alginate as a bioink in 3D printers aimed at the production of three-dimensional organotypical structures in order to, in the future, be able to replace injured tissues.

Natural polymers exhibit a relatively low mechanical strength compared to synthetic polymers. Thus, synthetic polymer cross-linking or blending improve natural polymer mechanical properties, although their biocompatibility is somewhat affected [135]. Therefore, many studies combine both types of polymers to improve polymeric matrix characteristics aiming at microencapsulation processes. Table 1 lists some studies employing natural polymers alone, with structural modifications or in combination with synthetic polymers in cell therapy.

Although the aforementioned findings are promising concerning tissue repair, further research is required in the search for a biomaterial more similar to matrix components.

**Table 1 polymers-14-02641-t001:** Biopolymers currently employed in cell therapy.

Biopolymer	Polymer Type	*In vitro* Tests	*In vivo* Test	Disease/ Disorder	Reference
Chitosan	Porcine collagen-chitosan	HUVECs cell survival and proliferation increases.	Improved angiogenesis and proliferation. Higher cell infiltration near the endothelial implant.	Subcutaneal implantation in CD1 mice.	[43]
	Chitosan/β-GP gel with DFO	hMSCs proliferation and survival induction; successful DFO release in HUVECs co-culture, increasing VEGF.	Not applicable.	Methodology protocol for CLI.	[44]
	Acidic chitosan	Induction rMSCs and hMSCs survival and proliferation. HUVEC cell growth by co-culture of encapsulated MSC.	Encapsulated Celastrol-treated cells induced increased vessel density, although the microcapsule induced an inflammatory response surrounding the implant with polynuclear cells and lymphocytes, as well as granulation tissue. Sprague Dawley rats.	Methodology protocol for CVD.	[46]
	CS-IGF1C	Increased ADSC proliferation and survival; hydrogel cell cryo-protection.	Engraftment enhancement and angiogenic induction.	Acute kidney injury in FVB mice.	[45]
	HBC-RGD hydrogel	Suitable BSA adsorption; BMSC viability and proliferation improvements.	Decreased keloid fibroblasts in *ex vivo* biopsies.	Keloid biopsy.	[48]
	CS—90% deacetylation	BMSC cell proliferation and survival; HUVEC pyroptosis suppression.	Improved engraftment in MI; decreased inflammatory response by cytokines (e.g., IL-6, TNF-α, IL-18) and caspases-11 and -1.	Acute myocardial infarction in FVB-Fluc/GFP mice.	[47]
Chitin	CMCH	Proliferation induction (HeLa and COS-7 cells). Unwanted precipitation of COS-7 cells.	Subcutaneal injection in C57BL/6J. No inflammation or cell death, suggesting a suitable milieu for cell viability.	Not applicable.	[49]
Alginate	Sodium alginate	Survival and differentiation induction of hMSCs into IPC cells with insulin production.	Greater insulin levels in male Swiss mice induced by a 50 mg/kg streptozotocin injection and glucose blood normalization.	Methodology protocol for diabetes.	[109]
	Alginate-GC	Increased survival and time-dependent insulin release in pancreatic islets from piglets, decreasing on the 32nd day.	Peritoneal injection in CD1 mice with HMW/LMW-GC-Alginate pancreatic islet encapsulation. Fibrotic response induction related to acrylate groups in the microbeads.	Methodology protocol addressed to diabetes.	[110]
	Collagen-Dextran sulfate-agarose	Encapsulated BMSCs or fibroblasts increased VEGF production and mixture-dependent cell differentiation and viability. Col-Fb-DxS100 exhibited better results.	Environment healing around the microcapsules with the presence of macrophages M1 (biomaterial phagocytosis) and M2 (anti-inflammatory milieu). Np fibrotic response induction.	Wistar rats—Myocardial ischemia model.	[136]
	Alginate core-shell microcapsule	rMSCs survival and differentiation induction.	Improvement in cardiac function and MSC migration into cardiac ischemic tissue. Sprague Dawley rats presenting induced MI.	Methodology protocol addressed to MI.	[119]
	Ultrapure alginate, low viscosity and high guloronic	Safety and metabolic function improvements in human hepatocyte allotransplantation.	Improvement of the liver and in metabolic function and no inflammatory response, although granulomatose inflammation was observed in the patients with fully recovered liver function.	Children with acute liver failure.	[115]
	APA microcapsule	Survival maintenance of intracardially cardiosphere-derived injected cells.	No difference was observed between the control and experimental groups, although an immune response was observed around the capsule.	Pigs with induced MI.	[120]
Collagen	Collagen-HA	Survival and metabolic function increases following hASC encapsulated administration.	Increased cell migration to a porcine cornea culture from encapsulated hASCs.	Not applicable.	[90]
	Collagen-alginate	Encapsulated IPC cells significantly induced insulin levels.	Encapsulated IPC cells were transplanted intra-dermally, and glucose blood levels returned to normal after 4 weeks.	BALB/C-Diabetes mice model.	[91]
Hyaluronic acid	Dexa-CB-1[6]/RA-DAH-HA	eMSC cell survival and function.	Survival after 60 days and IL-12M production, inducing tumor growth decreases. SKH1-E hairless mice and C57BL/6J tumor induced by the subcutaneous injection of B16F10 melanoma cells.	Tumor growth.	[67]
	MAP-HA coacervate	Maintenance of encapsulated rASCs survival and proliferation.	Favorable stem cell niche replacement from rASCs encapsulated by employing coacervate methods. In addition, increased VEGF and FGF2 production and platelet adhesion were noted following subcutaneal rat injections.	Methodology protocol for vessel impairment.	[68]
	Polyethylene glycol diacrylate-the ME-HA hydrogel microsphere	Maintenance of canine islet cell viability.	The microsphere attached the peritoneal wall; however, the xenotransplantation induces glucose blood normalization in NOD/SCID mice.	Methodology protocol for diabetes disorders.	[69]

Note: HUVECs: human umbilical vein endothelial cells; DFO: Desferrioxamine; β-GP: β—glycerophosphate; hMSCs: human mesenchymal stem cells; CLI: Critical limb ischemia; VEGF: Vessel endothelial growth factor; CS-NO: Chitosan-Nitric oxide; EC: Endothelial cells; SMC: Smooth muscle cells; T1DM: Type 1- diabetes mellitus; ADSCs: Adipose derived stem cells; HBC: Hydroxybutyl chitosan; BMSCs: Bone marrow stem cells; MI: Myocardia infarction; RGD: Arginine-Glycine-Aspartate-like; BSA: Bovine serum albumin; MAP: Mussel adhesive proteins; HMW: High molecular weight; APA: alginate-poly-l-lysine-alginate.

## 3. Cell Encapsulation Methods for Cell Therapy

Microcapsule performances are largely dictated by the physicochemical properties of the materials and the preparation techniques employed. Many preparation techniques using natural or synthetic polymers have been reported [137]. Not only are very reproducible methods needed for the preparation of devices with very precise parameters (permeability, size, surface), but these methods should additionally support cell integrity and viability during the encapsulation process and after implantation. Finally, the preparation method must ensure an adequate flux across the capsule membrane for cell survival and functions as well as long-term biocompatibility with host tissues [138].

Some traditional methods of microencapsulation are already well described and applied in cell trapping directed to cell therapy. Dripping technologies, i.e., forming droplets by extrusion through a nozzle, are still the main methods for cell encapsulation, which include simple dripping methods and electro- or vibration-assisted methods [139]. The methodology most often used in the process of obtaining the microcapsules is the dropwise regime with Coaxial air flow, which has an air flow system coaxially around the needle with the objective of decreasing the size of capsules [140,141].

However, despite numerous promising pre-clinical results in therapies using cell microencapsulation, at the present time, each method needs further improvements before reaching the proposed clinical phase. For this, some works have been studying modifications in the preexisting techniques with the objective of making advances towards the improvement of the production of encapsulated cells for the study of the treatment of different diseases [138]. One of the obstacles to be faced in the technique is the maintenance of cell viability. For that purpose, Qing-Quan Liao and colleagues described a cell encapsulation method in biocompatible sodium alginate droplets enveloped in oil droplets to form double emulsions. Ca^2+^ ions in the outer aqueous phase were diffused through the oil and introduced the gelation of sodium alginate to form cell encapsulation. This indirect gelation process and the spontaneous detachment of oil helped to keep cell viability [142]. Furthermore, with the aim of protecting the microencapsulated cells of the host’s immune system after transplantation, it has been shown that surface modification from the microcapsule or the immobilization and release of immunomodulatory molecules might also be considered as an effective strategy to suppress the host’s immune response upon implantation [143,144]. Thus, an immunosuppressant such as FK506, for example, can be incorporated into the outer layer of the alginate beads, using electrosprayed poly-ε-caprolactone core–shell nanoparticles with prolonged release profiles. This multi-layer encapsulation technology ensured the protection of cells from the central nervous system (CNS) and the protection of implanted cells from host immune responses whilst providing permeability to nutrients and released therapeutic molecules [144].

Therefore, the improvement of microcapsule production methodologies has been fundamental for the implementation of this approach as a therapeutic alternative.

## 4. Conclusions

The encapsulation of living mammalian cells within a semi-permeable hydrogel matrix is an attractive procedure for the development of new therapies, such as xenotransplantation, promoting an immunoprotective barrier to transplanted cells and also comprising a promising alternative to assist in the shortage of cellular and tissue sources in the treatment of various diseases. Many gaps still remain to be addressed in relation to cell therapy with encapsulated cells, such as their immunogenicity, the in situ production of soluble factors and tissue responses. Thus, it is important to highlight the need for the improvement of these techniques and their application mechanisms. This review therefore addressed the use of natural polymers and their importance for microencapsulation aimed at cell therapy, providing an overview of their current therapeutic applications.

## Figures and Tables

**Figure 1 polymers-14-02641-f001:**
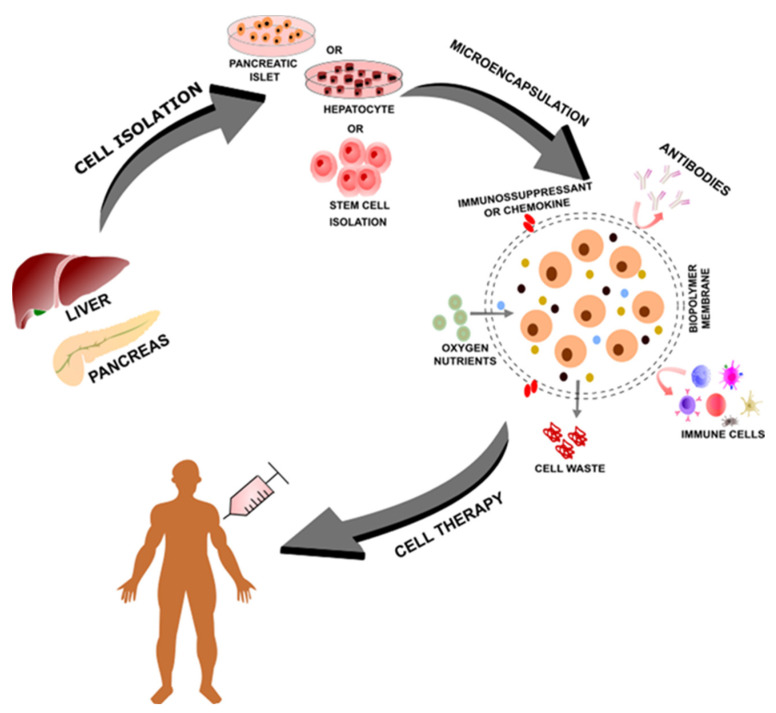
Schematic figure indicating cell microencapsulation. Cells isolated from organs, such as the liver and pancreas, or isolated stem cells may be encapsulated into a semipermeable biopolymer membrane, allowing for nutrient exchanges between the extra- and intracellular environments. This bidirectional link enables the entry of oxygen and growth factors and the exit of cell waste. Furthermore, the biopolymer membrane allows for decreased immunogenicity, with the inhibition of immune cell induction and antibody responses.

**Figure 2 polymers-14-02641-f002:**
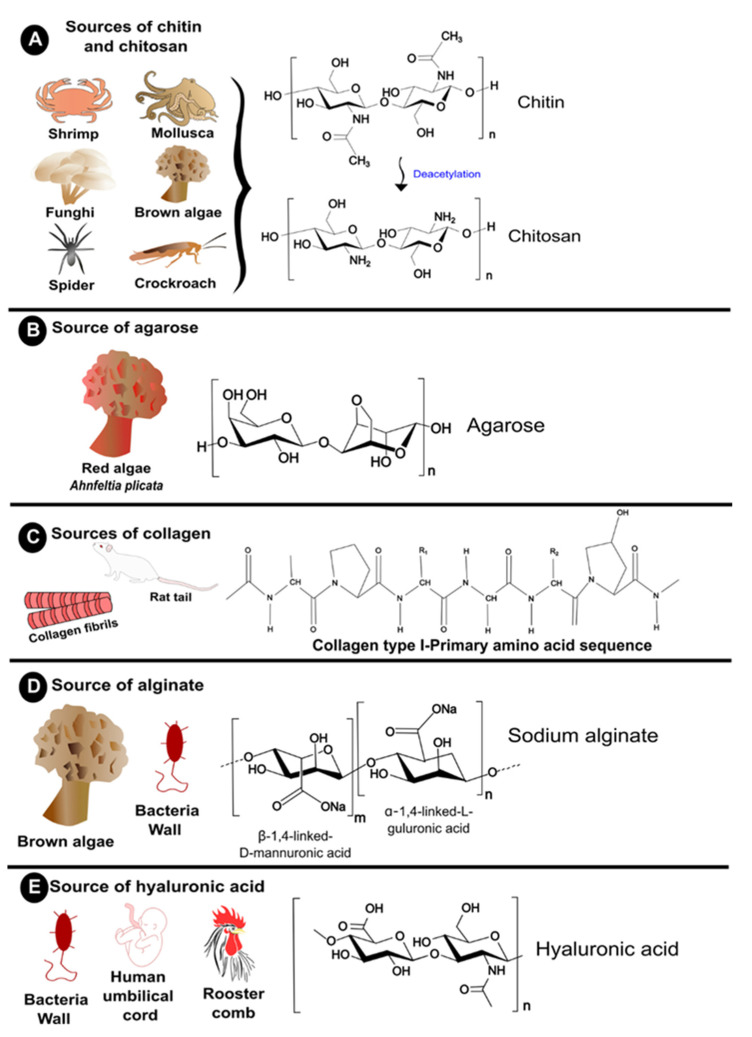
Schematic figure concerning the main source of the most frequently employed biopolymers for cells microencapsulation and their polymer structures. (**A**) Chitin and chitosan. (**B**) Agarose. (**C**) Collagen. (**D**) Alginate. (**E**) Hyaluronic acid.

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
