# Peer review of "Natural Biopolymers as Additional Tools for Cell Microencapsulation Applied to Cellular Therapy"

_polymers, 2022, doi:10.3390/polym14132641_

Round 1
Reviewer 1 Report
The authors should focus on cell therapy or change the title to better fit the content of the manuscript. That is, Table 1 should be elaborated.
The state of the art of cell encapsulation methods for cell therapy should be introduced and discussed carefully. In fact, in addition to materials, the vehicle for cell delivery is of importance for successful cell therapy. Consider to discuss cell encapsulation methods and cite, for example, DOI: 10.1039/D1RA08563H, DOI:10.1016/j.sna.2018.06.006, DOI: 10.1016/j.carbpol.2021.118262
English usage must be improved. The "additional" in the title of the manuscript is misspelled as addditional. There are many other grammar errors littered throughout the manuscript.
Reviewer 2 Report
The review entitled: Natural Biopolymers as Additional Tools for Cell Microen-capsulation in Cellular Therapy, it is an interesting article. In this review, the authors focused the attention on the description/discussion of natural biopolymers applied as Additional Tools for Cell Microen-capsulation in Cellular Therapy.
I suggest the publication of this review in polymers mdpi journal after minor revision.
Title: Addditional
Captions of figures: The captions of different figures are to long. The authors are invited to reduce the captions and are invited to describe the figures in the review.
The authors are invited (if they find the materials) to introduce a paragraph focusing the attention on the products present in commerce.
Round 2
Reviewer 1 Report
The manuscript is more get to the point in the current form.